

# Enhancing skin lesion classification: a CNN approach with human baseline comparison

Deep Ajabani[1,*], Zaffar Ahmed Shaikh[2,3,*], Amr Yousef[4,5], Karar Ali[6] and Marwan A. Albahar[7]

[1] Source InfoTech Inc., Loganville, Georgia, United States
[2] Department of Computer Science and Information Technology, Benazir Bhutto Shaheed University Lyari, Karachi, Pakistan
[3] School of Engineering, École Polytechnique Federale de Lausanne, Lausanne, Switzerland
[4] Electrical Engineering Department, University of Business and Technology, Jeddah, Saudi Arabia
[5] Engineering Mathematics Department, Alexandria University, Alexandria, Egypt
[6] VentureDive Pvt. Limited, Karachi, Pakistan
[7] College of Engineering and Computing in Al-Lith, Umm Al-Qura University, Makkah, Saudi Arabia
* These authors contributed equally to this work.

Corresponding authors
Zaffar Ahmed Shaikh,
zashaikh@bbsul.edu.pk
Amr Yousef, a.yousef@ubt.edu.sa

## ABSTRACT

This study presents an augmented hybrid approach for improving the diagnosis of malignant skin lesions by combining convolutional neural network (CNN) predictions with selective human interventions based on prediction confidence. The algorithm retains high-confidence CNN predictions while replacing low-confidence outputs with expert human assessments to enhance diagnostic accuracy. A CNN model utilizing the EfficientNetB3 backbone is trained on datasets from the ISIC-2019 and ISIC-2020 SIIM-ISIC melanoma classification challenges and evaluated on a 150-image test set. The model's predictions are compared against assessments from 69 experienced medical professionals. Performance is assessed using receiver operating characteristic (ROC) curves and area under curve (AUC) metrics, alongside an analysis of human resource costs. The baseline CNN achieves an AUC of 0.822, slightly below the performance of human experts. However, the augmented hybrid approach improves the true positive rate to 0.782 and reduces the false positive rate to 0.182, delivering better diagnostic performance with minimal human involvement. This approach offers a scalable, resource-efficient solution to address variability in medical image analysis, effectively harnessing the complementary strengths of expert humans and CNNs.

# INTRODUCTION

Early and precise detection of skin lesions is essential for effective treatment and improved patient outcomes (*Houssein et al., 2024*; *Ali et al., 2022*). Despite advancements in medical imaging technologies, significant challenges remain in achieving high diagnostic accuracy and efficiency in real-world clinical environments (*Jackson et al., 2025*; *Esteva et al., 2019*). Skin cancer stands out as a prevalent and aggressive cancer type, impacting over five

million individuals annually around the world (*Liu et al., 2020*; *Kurvers et al., 2019*). Swift and accurate diagnosis is key to effective treatment, prompting substantial investment in refining diagnostic tools (*Kiziloluk et al., 2024*; *Shaikh, 2009*).

Existing literature on skin lesion classification highlights three primary research categories, each addressing distinct challenges in medical diagnostics (*Ahmad et al., 2024*; *Zalaudek et al., 2006*). The first focuses on enhancing human decision-making processes to bolster accuracy (*Yang et al., 2024*; *Kurvers et al., 2021a*, *2021b*). This involves consolidating expert opinions (*Liu et al., 2024*) or devising diagnostic techniques that yield heightened precision (*Combalia et al., 2019*; *Brinker et al., 2018*). However, while these efforts have improved diagnostic accuracy, they are resource-intensive, requiring multiple experts for consensus (*Hernández-Pérez et al., 2024*; *Kousar et al., 2024*).

The second category revolves around artificial intelligence (AI) (*Hosseinzadeh et al., 2024 Sadeghi et al., 2024*), particularly advancements in convolutional neural networks (CNN) for malignancy classification in medical images (*Chatterjee, Gil & Byun, 2024*; *Shaikh et al., 2022*). Several studies have demonstrated that CNNs outperform individual clinicians in skin cancer detection tasks, underscoring their potential to enhance diagnostic accuracy (*Chen et al., 2024*; *Bingol & Alatas, 2021*; *Rezvantalab, Safigholi & Karimijeshni, 2018*). Yet, the generalizability of AI systems remains a challenge, as most studies are conducted under controlled conditions, making it difficult to translate these results directly into clinical settings (*Haenssle et al., 2020*; *Hekler et al., 2019*; *Haenssle et al., 2018*).

The third category explores hybrid models that integrate the first two categories of human expertise with AI predictions, to achieve superior performance (*Tschandl et al., 2019*; *Brinker et al., 2018*; *Shaikh & Lashari, 2017*). For example, studies have shown that ensemble models combining the opinions of multiple clinicians tend to outperform individual diagnoses, provided that their performance levels are comparable (*Kurvers et al., 2019*). However, when performance varies significantly among the experts, ensemble methods may underperform compared to the top-performing individual clinician (*Chatterjee, Gil & Byun, 2024*). This variability highlights the complexity of human-machine collaboration, where balance and structure are essential for optimal results (*Jafar et al., 2024*; *Shaikh et al., 2022*; *Han et al., 2018*).

Similarly, *Haenssle et al. (2020)*, *Marchetti et al. (2020)*, *Brinker et al. (2019)*, *Kurvers et al. (2019)*, *Mahbod et al. (2019)*, *Haenssle et al. (2018)*, and *Esteva et al. (2017)* used CNNs and compared them against human experts in classifying skin lesions through image analysis. All these studies consistently showcased the superiority of machines over humans in this domain. This assertion supports multiple other studies (*Akram et al., 2025*; *Ali et al., 2022*; *Marchetti et al., 2020*). A comprehensive analysis by *Haggenmüller et al. (2021)* concluded that all 19 studies they covered in their comparison demonstrated superior or at least equivalent performance of CNN-based classifiers compared with clinicians.

Despite these advancements, there remains a critical gap in practical and scalable solutions that combine human and machine intelligence while considering real-world

constraints (*Park et al., 2023*; *Brady & Neri, 2020*; *Topol, 2019*). Many of the comparative studies mentioned above were conducted under controlled conditions. Furthermore, as highlighted by *Houssein et al. (2024)*, *Nugroho, Ardiyanto & Nugroho (2023)*, *Cassidy et al. (2022)*, and *Haenssle et al. (2020)*, many comparative studies place clinicians in an unfamiliar setting by requiring them to make predictions solely from images, without access to other clinical information. This study addresses this gap by proposing a hybrid algorithm, coined the "augmented hybrid approach". This approach aims to optimize diagnostic performance through selective collaboration between human experts and AI models. It offers an economically viable solution with the potential to improve outcomes and save lives (*Tao & Alatas, 2024*; *Shaikh et al., 2021a*; *Mahbod et al., 2019*).

This augmented hybrid approach extends the concept outlined in studies by *Pirrera & Giansanti (2023)*, *Brady & Neri (2020)*, *Topol (2019)*, *Han et al. (2018)*, which enhances predictive capabilities by providing clinicians with CNN prediction scores as supplementary information.

In this scenario, when a human expert is uncertain about a diagnosis and the algorithm demonstrates high confidence, the expert defers to CNN's prediction, leading to more informed decision-making. This methodology assumes that both human and algorithmic performances align with their confidence in their respective predictions. However, humans often struggle to accurately estimate their confidence, leading to suboptimal use of AI-generated insights (*Akhund et al., 2024a*; *Ha, Liu & Liu, 2020*; *GitHub, 2024a*). As a result, augmented intelligence may not reach its full potential if CNN predictions are underutilized by humans (*Ali et al., 2023*; *Shaikh, 2018*). To address this, our study proposes an algorithmic framework that reverses the traditional approach of relying on human confidence. We use CNN prediction confidence as a proxy for certainty. By replacing human responses with CNN predictions in cases where the network exhibits high uncertainty, we demonstrate a significant improvement in overall performance. This approach is based on prior research, focusing on CNN prediction certainty rather than human confidence estimation (*Ahmad et al., 2024*; *Deotte, 2020*).

This augmented hybrid approach aims to enhance diagnostic precision and the efficient utilization of human resources. Combining the strengths of CNNs with the expertise of medical professionals offers a cost-effective solution to reduce clinician workload and ultimately elevate the quality of treatment (*De, Mishra & Chang, 2024*; *Saeed et al., 2024*; *Dayananda et al., 2023*; *Secinaro et al., 2021*).

For performance evaluation, this research develops a hybrid algorithm that employs the EfficientNetB3 backbone for CNN training, utilizing the ISIC-2019 and ISIC-2020 datasets (*ISIC, 2024*)—the two comprehensive, widely popular, and open-access datasets used in the SIIM-ISIC (Society for Imaging Informatics in Medicine—International Skin Imaging Collaboration) melanoma classification challenges (*Saghir, Singh & Hasan, 2024*; *ISIC, 2024*; *Tan & Le, 2019*; *Codella et al., 2018*; *Gutman et al., 2016*).

By comparing the performance of this hybrid model with both the baseline CNN and human experts, we demonstrate the feasibility and benefits of integrating AI into

dermatological diagnostics of skin lesion classification (*Farea et al., 2024*; *Gholizadeh, Rokni & Babaei, 2024*; *Pirrera & Giansanti, 2023*; *GitHub, 2024a*, *2024b*; *Kassem et al., 2021*; *Kassani & Kassani, 2019*; *Han, Mao & Dally, 2015*). This study suggests a hybrid approach that combines CNN predictions with human expertise, potentially improving diagnostic performance by mitigating human errors and machine inaccuracies.

The ISIC-2019 and ISIC-2020 datasets (*ISIC, 2024*) provide benchmarks for training and evaluation, ensuring alignment with current practices and validated methodologies (*Jackson et al., 2025*; *Gouda et al., 2022*; *Adegun & Viriri, 2021*). These datasets also highlight the effectiveness of EfficientNet models, which are recognized for their computational efficiency (*Tan et al., 2024*; *Debelee, 2023*; *Tan & Le, 2019*). In this study, data augmentation techniques are employed to further enhance the performance and generalization of the EfficientNet model across diverse medical images (*Kumar et al., 2024*; *Batool & Byun, 2023*; *Hekler et al., 2020*). This adaptability is essential for real-world clinical settings, where imaging conditions vary significantly (*Shaikh et al., 2022*; *Shorten & Khoshgoftaar, 2019*). Additionally, we collected expert evaluations from 170 medical professionals, primarily dermatologists, to ensure the reliability and robustness of the comparative analysis. From this pool, 69 participants with extensive experience in dermoscopy were selected based on stringent inclusion criteria, reflecting real-world clinical scenarios and providing meaningful insights into the proposed augmented hybrid model's effectiveness. Ultimately, this work contributes to advancing medical imaging by offering a scalable, efficient, and reliable framework for AI-assisted diagnosis (*Akhund et al., 2024b*; *Alam et al., 2022*; *Khamparia et al., 2021*; *Mahbod et al., 2019*; *Han, Mao & Dally, 2015*).

The article is as follows; related studies are discussed in the following section. The section provides an analysis of data, including preprocessing, model selection, computing infrastructure, data evaluation, training data, and train-test split. In "Empirical Study" we conduct research, which involves choosing the model and assessing the metrics. "Training a CNN" covers the training of CNN detailing our attempt and the challenges faced. Our baseline experiments are outlined in "Baseline Experiments". The hybrid algorithm is explained in "Hybrid Algorithms". We discuss the limitations of our study in "Study Limitations" and conclude the article in "Conclusion" by presenting our study results and suggesting areas for further research.

# RELATED WORKS

This section aims to represent the works related to the present objectives of the study. There are three primary research directions.

## Aggregation of expert opinions

The first research direction focuses on aggregating opinions, which proves beneficial when the ensemble members exhibit similar performance levels (*Hosseinzadeh et al., 2024*; *Freeman et al., 2020*; *Kay et al., 2018*; *Lane et al., 2017*; *Shaikh & Khoja, 2012*). This approach stands to enhance clinical performance but demands substantial resources, as a

majority consensus requires soliciting input from multiple clinicians for each case (*Wubineh, Deriba & Woldeyohannis, 2024*; *Hussain et al., 2023*; *Bejnordi et al., 2017*).

## CNNs *vs.* human experts

The second research direction asserts that CNNs demonstrate comparable or superior proficiency to humans in categorizing skin lesion malignancies (*Navarrete-Dechent, Liopyris & Marchetti, 2020*). However, further inquiry remains imperative in this domain as current algorithms have yet to reach a level where they can replace human classification (*Chatterjee, Gil & Byun, 2024*; *Esteva et al., 2017*). Consequently, investigating hybrid methodologies represents a logical progression in research (*Hasan et al., 2022*). Various hybrid approaches have been presented, demonstrating superior performance compared to individual human or machine capabilities (*Bozkurt, 2023*; *Goceri, 2020*). Despite the potential of hybrid intelligence to save lives (*Brinker et al., 2019*; *Liu et al., 2020*; *Tschandl et al., 2019*; *Ketkar & Santana, 2017*), a critical gap in the literature pertains to the expenses associated with such methods (*Wubineh, Deriba & Woldeyohannis, 2024*; *Chollet & Chollet, 2021*), which can make even highly effective approaches impractical for real-world implementation (*Topol, 2019*). For instance, the study by *Hekler et al. (2019)* involved the use of multiple clinicians to assess a single image, which would be infeasible in clinical practice.

## Hybrid methodologies and the challenges of AI

Similar findings were observed by *Houssein et al. (2024)*, *Koçak et al. (2024)*, *Secinaro et al. (2021)*, *Tschandl et al. (2020)*, who highlighted that less experienced clinicians benefited the most from computer predictions. However, they cautioned that faulty AI could mislead clinicians across all experience levels, posing a threat to the usability of hybrid approaches (*Kaluarachchi, Reis & Nanayakkara, 2021*; *Han et al., 2018*; *Chollet & Chollet, 2021*). Blindly trusting AI decisions could undermine the superior performance achieved through human-machine collaboration, as demonstrated in these studies (*Brinker et al., 2018*; *Goodfellow, 2016*; *Shaikh & Khoja, 2011*). For example, *Ha, Liu & Liu (2020)*, *Marchetti et al. (2020)* pursued a systematic approach where human participants rated their confidence in each image. By supplementing low-confidence ratings with machine-generated predictions, performance improved, particularly for medical residents, but less so for professional dermatologists. Although this method mitigates the risk of over-reliance on CNN predictions, it remains susceptible to flawed AI (*Gómez-Carmona et al., 2024*; *Krakowski et al., 2024*; *Keerthana, Venugopal & Nath, 2023*).

## Ensemble approaches for skin lesion classification

The third research direction is the use of ensemble approaches (*Aboulmira et al., 2025*; *Hekler et al., 2019*). By aggregating predictions using a boosting algorithm, *Gulli & Pal (2017)* found that combining human and computer decisions yielded the best results for multiclass and binary lesion classification. This ensemble approach prevents human reliance solely on machine-generated results and appears to be a more feasible real-world implementation (*Yang et al., 2024*; *Reddy et al., 2024*). However, comprehensive

comparisons of these approaches are still lacking (*Ali et al., 2023*). In the examination of existing literature (*De, Mishra & Chang, 2024*; *Kumar et al., 2024*; *Tschandl et al., 2020*; *Shaikh & Khoja, 2014*), the majority of studies focused on skin lesion classification relied on the utilization of one or more pre-trained networks as a foundational framework for their models.

In Table 1 we present a well-rounded comparative analysis of recent advancements, hybrid models, and interpretability-focused essential and recent studies in skin lesion classification literature that align closely with the objectives and key themes of this study.

## DATA

This study utilizes three distinct datasets for analysis: the original BCN20000 dataset of the three-point checklist of dermoscopy (*Combalia et al., 2019*) and the ISIC-2019 and ISIC-2020 datasets (*ISIC, 2024*). The BCN20000 dataset comprises 165 images alongside corresponding human responses. This dataset is deemed suitable for constructing and validating the ensemble technique (*Freeman et al., 2020*; *Shaikh et al., 2019*), serving as the evaluation dataset herein. However, its size proves insufficient for adequate CNN training (*Saeed et al., 2024*; *Batool & Byun, 2023*; *Sun et al., 2023*). Therefore, substantially larger datasets, ISIC-2019 and ISIC-2020, denoted as the training datasets, were employed for this purpose. The subsequent sections delineate the procedures involved in gathering and preprocessing these datasets, with priority given to an initial overview of the evaluation dataset.

### Data preprocessing

During the research, preparing the data was crucial to train CNN models efficiently (*Pérez & Ventura, 2022*; *Marchetti et al., 2020*). These initial steps were necessary to get the dataset ready, for training and evaluation guaranteeing that the CNNs could learn well and provide predictions (*Koçak et al., 2024*; *Hekler et al., 2020*; *Brinker et al., 2019*).

All images were uniformly downsized to a resolution of $256 \times 384$ pixels. This decision balanced the need for detail preservation and computational efficiency, as higher resolutions would increase training time without guaranteeing better performance (*Aboulmira et al., 2025*; *Ali et al., 2021*). To enhance the training dataset, an ImageDataGenerator was used to perform various augmentation techniques such as rotation, zoom, and horizontal flipping (*Bozkurt, 2023*; *Shaikh et al., 2021b*; *Goceri, 2020*). This approach helped in creating a more robust model by exposing it to a variety of image transformations, thereby improving generalization (*Hekler et al., 2020*; *Han et al., 2018*).

During the training and validation phases, we opted for a batch size of 64 to ensure a mix of classes, in each batch which is important for addressing class imbalances (*Aboulmira et al., 2025*). When it came to testing, we used a batch size of 1 to maintain the image order and ensure alignment between predictions and actual data (*Shaikh et al., 2024*; *Shaikh & Khoja, 2014*). The images were normalized to standardize the input data aiding in speeding up the CNNs learning process by ensuring that the data distribution has an average of zero and a standard deviation of one (*Chollet & Chollet, 2021*; *Brinker et al., 2019*).

**Table 1 Summary of pre-trained models.**

| Reference | Network(s) used |
| --- | --- |
| *Brinker et al. (2019)* | ResNet50 |
| *Abdelrahman & Viriri (2023)* | EfficientNetB3, EfficientNetB4, EfficientNetB5, EfficientNetB6, EfficientNetB7, Se-ResNext101, ResNest101 |
| *Han et al. (2018)* | ResNet-152 |
| *Hekler et al. (2020)* | ResNet50 |
| *Haenssle et al. (2018)* | InceptionV4 |
| *Haenssle et al. (2020)* | MoleAnalyzerPro |
| *Li et al. (2020)* | Custom |
| *Esteva et al. (2017)* | Inception V3 |
| *Haggenmüller et al. (2021)* | AlexNet, VGG16, VGG19, GoogleNet, ResNet-50, ResNet-101, ResNet-152, Inception-V3, Inception-V4, DenseNets, SeNets, PolyNets |
| *Han et al. (2020)* | SeNet, Se-ResNet50, VGG19 |
| *Tschandl et al. (2020)* | ResNet34 |

## Justification of model types and selection method

### Model types

This study employed EfficientNetB3 models for skin lesion classification, chosen for their balance of high performance and computational efficiency (*Aboulmira et al., 2025; Kassem et al., 2021; Tan & Le, 2019*). The justification for selecting EfficientNetB3 and the model types used are based on several key factors.

### Performance and efficiency

EfficientNets have shown better results than networks having a similar parameter count, as seen in *Hasan et al. (2021), Huang et al. (2022)*. They deliver outcomes in tasks, like image classification while being computationally effective making them a budget-friendly option for this research (*Aboulmira et al., 2025; Ali et al., 2022; Zalaudek et al., 2006*). EfficientNet model EfficientNetB3 uses parameters compared to various conventional CNN designs (*Alhichri et al., 2021; Li et al., 2018*). This efficiency helps decrease the workload during training, which is essential considering the limited computing resources accessible for this study (*Akhund et al., 2024a; Ha, Liu & Liu, 2020; Hekler et al., 2020*).

### Demonstrated effectiveness

EfficientNets, especially the EfficientNetB3 variant, were prominently used in high-ranking submissions of the 2019 and 2020 SIIM-ISIC melanoma classification challenges (*ISIC, 2024*). This track record of success in similar dermatological imaging tasks is also seen in *Reddy et al. (2024), Gouda et al. (2022)*, and *Feng et al. (2022)*, underscores their suitability for skin lesion analysis.

To prevent overfitting, the study employed various regularization techniques, including data augmentation and dropout layers (*Marchetti et al., 2020; Srivastava et al., 2014; Argenziano et al., 2003*). EfficientNetB3's design allows for the integration of these techniques (*Aboulmira et al., 2025; Kim & Bae, 2020*), further enhancing its performance on the validation set and ensuring better generalization to new data (*Salman & Liu, 2019*).

## Selection method

The selection method for determining the most suitable model involved the study exploring a grid of models with variations in label encoding (binary/multiclass), model head capacity (shallow/deep), and dropout layer strength (0/0.2/0.4/0.6) (*Jackson et al., 2025*; *Bergstra & Bengio, 2012*). This comprehensive grid search identified the optimal combination of these parameters to maximize model performance (*Tuba et al., 2021*). Early stopping was implemented to terminate training when no improvement in validation loss was observed for seven consecutive epochs (*Tuba et al., 2021*; *Yu, Song & Ren, 2013*). This approach ensured that models did not overfit, and training resources were used efficiently (*Saghir, Singh & Hasan, 2024*; *LeCun et al., 1998*; *Smith, 2017*; *Dietterich, 1995*).

To expedite the training process, models were trained in parallel and grouped by target label and model capacity (*Nugroho, Ardiyanto & Nugroho, 2023*; *Huang et al., 2017*). This strategy significantly reduced the total training time, allowing for a thorough exploration of the model grid within a feasible timeframe (*Akram et al., 2025*; *Brinker et al., 2019*). After optimizing models on the validation set, their performance was evaluated on an independent test set (*Huang et al., 2022*; *Abadi et al., 2016*). This final evaluation step ensured that the selected models generalize well to new, unseen data (*Loshchilov & Hutter, 2016*).

In summary, EfficientNetB3 was chosen for its demonstrated performance and efficiency (*Chollet & Chollet, 2021*; *Tschandl et al., 2019*), with the selection method ensuring robust and generalizable model performance through a systematic and resource-efficient training process (*Hosseinzadeh et al., 2024*).

## Computing infrastructure

We required enormous computing power to conduct this research because of the nature and large scale of the deep learning models and datasets utilized (*NVIDIA Corporation, 2020*).

The research made use of Linux-based operating systems for their reliability and efficiency (*Gulshan et al., 2016*), in handling tasks and compatibility with various deep learning frameworks (*Khalil et al., 2023*). Training learning models, such as CNNs like EfficientNet demand GPU power (*Jouppi et al., 2017*). The study employed high-performance NVIDIA GPUs optimized for learning tasks (*Kang & Tian, 2018*; *NVIDIA Corporation, 2020*). While the research considered solutions utilizing TPUs (Tensor Processing Units) known for their efficiency in tensor operations in networks, the practical implementation primarily depended on GPU resources due to their availability and infrastructure limitations (*Jouppi et al., 2017*; *Abadi et al., 2016*; *Nair & Hinton, 2010*). Multi-core CPUs played a role in preprocessing data and overseeing workflow management.

Handling large datasets such as BCN20000, ISIC-2019, and ISIC-2020 requires high-capacity storage solutions (*Ali et al., 2022*; *Nguyen et al., 2019*; *ISIC, 2024*). Fast SSDs were used to ensure quick data access and processing speeds (*Aboulmira et al., 2025*; *Kumar et al., 2024*). Training complex models and processing large datasets necessitated large amounts of RAM to handle data efficiently during training phases.

The main tool for developing and training models was TensorFlow (*Jang, 2025*; *Pang, Nijkamp & Wu, 2020*; *Dillon et al., 2017*; *Abadi et al., 2016*). This framework is widely supported on Linux (*Abadi et al., 2016*; *Nair & Hinton, 2010*). It provides a range of tools for creating and improving neural networks. Kaggle Notebooks were used for model testing (*Mostafavi Ghahfarokhi et al., 2024*; *Mukhlif et al., 2024*). To benefit from community-shared solutions, these notebooks offered an adaptable environment for conducting experiments with computing requirements. Various personalized scripts—for processing, enhancing, and assessing the data—were created to customize the workflows according to the study's needs (*Akhund et al., 2024b*; *Banachewicz & Massaron, 2022*; *Abadi et al., 2016*).

The computing setup relied on GPUs, ample memory, spacious storage, and reliable deep-learning software, on a Linux system to support smooth and productive model training and assessment procedures (*Haggenmüller et al., 2021*; *Nair & Hinton, 2010*).

## Evaluation data

### Image dataset description

The BCN20000 dataset comprised 165 images selected randomly from a larger collection of 2,621 images. The sole criteria for inclusion were adequate image quality and the presence of hemoglobin pigmentation in either the entire lesion or part thereof (*Mostafavi Ghahfarokhi et al., 2024*; *Combalia et al., 2019*; *Esteva et al., 2017*). Among these 165 images, 15 were allocated for training purposes, facilitating participant familiarity with the process of utilizing the three-point checklist for evaluation. The remaining subset of 150 images served as the basis for assessing performance in this study. This set of 150 images, along with the corresponding human evaluations, forms the core of the evaluation dataset employed in this research endeavor (*Rotemberg et al., 2021*; *Tschandl et al., 2020*). Specifically, the BCN20000 dataset offered the following components:

- JPEG (JPG) files with a resolution of 512 × 768 pixels for each image.
- Individual evaluations by participants utilizing the three-point checklist for each image.
- Ground truth data corresponding to each image.
- Metadata associated with the images.
- Metadata associated with the participating individuals

Each image in the BCN20000 dataset was presented at a resolution of 512 pixels in height and 768 pixels in width, thereby establishing an aspect ratio of 1:1.5 (*Hernández-Pérez et al., 2024*). To ensure consistency in training and testing processes with similar images, the training images referred to in the preceding section were adjusted to the same aspect ratio (*Tan et al., 2024*; *Combalia et al., 2019*).

### Image classification and ground truth

The investigation focuses on categorizing images through binary classification, distinguishing between a "benign" and a "malignant" category. The benign class represents a negative status denoting no cause for concern, while the malignant class signifies the presence of cancer (*Chatterjee, Gil & Byun, 2024*; *Marchetti et al., 2020*). Study participants

were not directed to categorize images as benign or malignant; instead, they assessed each image based on three distinct characteristics: asymmetry (about color and/or structure, not shape), atypical network (characterized by a pigment network displaying thick lines and irregular holes), and blue-white structures (indicating the presence of blue and/or white coloration within the lesion) (*Mostafavi Ghahfarokhi et al., 2024*; *Tschandl et al., 2019*). When two or more of these characteristics were identified, the lesion was classified as malignant. To establish malignancy scores for each participant and image, the responses for each criterion were converted into binary form, where 0 represented "not present" and 1 indicated "present". The analysis discerns the classification of images into negative or positive classes for each participant (*Jackson et al., 2025*; *Marchetti et al., 2020*). The ground truth of each image was established *via* histopathological examination (*Esteva et al., 2019*). Among the 165 images, 116 were benign instances and 49 were malignant instances, indicating an incidence rate of 29.7% (*Haenssle et al., 2018*). This rate notably exceeds that of the training images, which could potentially impact the outcomes (*Marchetti et al., 2020*). Accompanying each image is metadata detailing the subject of the image, encompassing age, sex, and the lesion's anatomical location, akin to the information provided in the ISIC-2019 and ISIC-2020 datasets (*ISIC, 2024*; *Tan & Le, 2019*). Nonetheless, the decision was made to exclude metadata from the model.

### Participant selection criteria

The evaluation encompassed 170 participants, who provided background information related to their professional roles and medical experience. This information included details such as:

- **Professional background:** Participants were primarily dermatologists or medical professionals with substantial expertise in skin lesion analysis.
- **Country of origin:** The participants came from various regions, ensuring geographic diversity. This may have helped capture a range of diagnostic approaches and perspectives.
- **Experience with dermoscopy:** Participants were asked to report their prior experience with dermoscopy, including whether they were routinely engaged in diagnosing skin lesions through dermoscopy in clinical practice.
- **Years of experience:** The number of years participants had been performing dermoscopies was recorded to ensure that only individuals with sufficient experience were included in the performance evaluation.
- **Frequency of yearly dermoscopies:** To further quantify their expertise, participants also reported how often they perform dermoscopies each year. This helped in distinguishing between frequent and less frequent users of the technique.

For the scope of this study, the inclusion was limited to participants who had completed at least 126 out of the 150 images in the evaluation dataset. This threshold was set to ensure a reliable assessment of each participant's performance. Based on this criterion, 69 experienced participants were selected for the analysis. This approach was designed to

reflect a real-world clinical scenario and to provide a meaningful comparison between human evaluators and the CNN model (*Lee et al., 2025*).

## Training data

A substantial volume of high-quality images is crucial to train a CNN effectively (*Ali et al., 2022*; *Rotemberg et al., 2021*). For instance, the ImageNet Challenge used 1.2 million images across 1,000 categories, averaging 1,200 images per category (*Hekler et al., 2019*; *Gutman et al., 2016*; *Deng et al., 2009*). However, gathering and verifying these images is labor-intensive, resulting in limited datasets for computer vision tasks (*Tan et al., 2024*). Exploring skin lesion datasets, the PH2 dataset provided only 200 images, insufficient for CNN training (*Gouda et al., 2022*). The SIIM-ISIC challenges between 2016 and 2020 saw a significant expansion (*ISIC, 2024*) as shown in Table 2.

The ISIC-2020 dataset included 33,126 images (*ISIC, 2024*; *Rotemberg et al., 2021*), curated from various sources after thorough quality checks. Despite the volume, the ISIC-2020 dataset had a low positive incidence rate of 1.76%, potentially causing imbalances in training. To address this, merging the BCN20000 dataset with a 17.85% incidence rate was necessary, providing more positive instances for better model learning. Both datasets were used for this reason. The inclusion of metadata (patient age, sex, lesion site) in skin lesion classification models has shown promise. Studies suggest that incorporating such data enhances diagnostic accuracy for clinicians (*Reddy et al., 2024*; *Haggenmüller et al., 2021*). However, some winning Kaggle models (*Lee et al., 2025*; *Ha, Liu & Liu, 2020*; *Lopez et al., 2017*) did not benefit from metadata fusion, opting for different strategies. *Haenssle et al. (2018)* also highlight the potential of metadata but suggest varying benefits depending on the model structure (*Naseri & Safaei, 2025*; *Codella et al., 2018*). Aligning labels for both datasets, as shown in Table 3, involved treating the "MEL" class as positive and others as negative. Yet, using multiclass labels poses a risk: it might improve performance but limit overall utility (*Shen et al., 2019*; *Deng et al., 2009*). To counter this, we experimented with training models using both binary and multiclass labels.

## Train-test split

Neural networks, due to their high capacity, are prone to overfitting, making it unsuitable to evaluate them on the same data used for training (*Abbas et al., 2025*; *Ophir et al., 1991*). Standard practice involves splitting the data into three sets: training, validation, and test sets. The training set is used to optimize the model's performance, but overfitting can occur if the model memorizes the training data (*Alotaibi & AlSaeed, 2025*; *Salman & Liu, 2019*). A validation set, derived from the training set, helps identify overfitting by monitoring the loss on both training and validation sets after each epoch (*Liu et al., 2025*; *LeCun et al., 1998*). Overfitting is detected when training loss decreases while validation loss increases (*Srivastava et al., 2014*). The validation set also determines when to stop training to avoid further overfitting (*Debelee, 2023*). However, models optimized excessively on the validation set risk "overfitting to the validation set", where some models perform better by chance rather than reflecting the true model. To mitigate this, a test set—a separate subset—is used to assess the final model's ability to generalize new, unseen data. After

**Table 2 ISIC competition in the year 2016–2020.**

| Year | # of images |
|---|---|
| 2016 | 900 |
| 2017 | 2,000 |
| 2018 | 12,609 |
| 2019 | 25,331 |
| 2020 | 33,126 |

**Table 3 Alignment procedure for the 2019 and 2020 ISIC data labels.**

| 2019 Diagnosis | 2020 Diagnosis | Target |
|---|---|---|
| NV | Nevus | NV |
| MEL | Melanoma | MEL |
| BCC | BCC | BCC |
| BKL | Seborrheic keratosis, lichenoid keratosis, solar lentigo, lentigo NOS | BKL |
| AK | | AK |
| SCC | | SCC |
| VASC | | VASC |
| DF | | DF |
| | Cafe-au-lait macule, atypical melanocytic proliferation, unknown | Unknown |

optimizing the validation set, the model's performance on the test set ensures its suitability for predicting novel images (*Aboulmira et al., 2025*; *Tan & Le, 2019*).

In this study, we employed a 15% validation split (*Liu et al., 2024*, *2020*; *Chollet & Chollet, 2021*). Due to differing incidence rates in the ISIC-2019 and ISIC-2020 datasets (*Hernández-Pérez et al., 2024*; *ISIC, 2024*; *Rotemberg et al., 2021*), a stratified train-validation split was implemented, using an 85-15 division for each dataset before combining them (*Naseri & Safaei, 2025*; *Ali et al., 2022*; *Deotte, 2020*). Table 4 shows the class distribution for these splits. The test set provided by the 2020 Kaggle competition was used directly, omitting the need for a custom test set. This test set of the ISIC-2020 dataset comprises 10,982 images without ground truth, preventing model-specific tuning (*ISIC, 2024*). Instead, Kaggle accepts model predictions for scoring, facilitating performance comparisons with competition participants (*ISIC, 2024*; *Ha, Liu & Liu, 2020*).

## EMPIRICAL STUDY

An analysis was conducted with more than 30 pre-trained networks available for selection. Table 1 comprehensively illustrates the prevalent use of ResNet networks as foundational structures for skin lesion analysis. However, the prominent submission in the 2020 SIIM-ISIC melanoma classification challenge predominantly leveraged EfficientNets to highlight the consistently superior performance of EfficientNets compared to other networks with similar parameter counts (*ISIC, 2024*; *Tan et al., 2024*). This suggests these networks might be more cost-effective due to their lower parameter count, demanding less

**Table 4 Class distribution in training and validation sets.**

| Category | Training share % | Validation share % |
|---|---|---|
| AK | 1.5 | 1.6 |
| BCC | 5.6 | 6.0 |
| BKL | 4.9 | 4.6 |
| DF | 0.4 | 0.4 |
| MEL | 8.8 | 8.6 |
| NV | 31.0 | 30.3 |
| SCC | 1.1 | 1.1 |
| Unknown | 46.3 | 47.1 |
| VASC | 0.4 | 0.4 |

computational resources during training (*Gouda et al., 2022*). A comprehensive review of pre-trained networks within the Keras package further substantiates the effectiveness of EfficientNets (*ISIC, 2024*). They exhibit creditable performance in the ImageNet Challenge while necessitating fewer parameters and reasonable training durations (*Houssein et al., 2024*; *Hekler et al., 2019*). Given constrained computational resources, opting for an EfficientNet seems a judicious choice for this study (*Ali et al., 2022*). Specifically, selecting the EfficientNet B3 variant aligns with the models utilized (*Hosseinzadeh et al., 2024*; *Tan & Le, 2019*; *Ophir et al., 1991*).

## Regularization

Deep neural networks, specifically Deep CNNs, exhibit extensive model capacity, making them susceptible to overfitting. Techniques employed to counter overfitting are known as regularization methods and encompass various approaches.

In this study, four key regularization techniques—data augmentation, dropout layers, capacity regulation, and weight regularization—are optimized following prior research (*Srivastava et al., 2014*; *Argenziano et al., 2003*). *Chollet & Chollet (2021)* suggests exploring model capacity until overfitting appears, followed by applying regularization methods to improve test performance. This process is iterative, time-consuming, and requires expertise from the data scientist (*Hekler et al., 2019*). Data augmentation prevents overfitting by altering input data, ensuring the model learns general features rather than specific images. This technique is crucial for small datasets and also improves generalization in larger datasets (*Goodfellow, 2016*). Rotation, zooming, and horizontal flipping are used in this study to introduce variations, helping the model handle novel orientations (*Hekler et al., 2019*; *Zalaudek et al., 2010*). Although shear is a common augmentation strategy, *Zalaudek et al. (2006)* highlight its potential to distort asymmetrical features, crucial for detecting malignant melanomas, which led to its exclusion. Based on prior literature observations (*Hernández-Pérez et al., 2024*; *Chollet & Chollet, 2021*; *Goodfellow, 2016*; *Srivastava et al., 2014*), data augmentation remains a vital part of regularization in all models.

Dropout layers, situated between hidden layers in a neural network, employ a binary mask to deactivate part of the activation, compelling subsequent layers to operate with incomplete information (*Hernández-Pérez et al., 2024*; *Srivastava et al., 2014*). This technique effectively converts the model into a correlated ensemble without multiple model instances (*Hekler et al., 2019*). Regulating the capacity of a neural network involves adjusting its depth and width, impacting the number of parameters. EfficientNets were developed by striking a balance between these dimensions, aiming for an optimal structure (*Mukhlif et al., 2024*; *Haggenmüller et al., 2021*; *Tan & Le, 2019*; *Dillon et al., 2017*; *Goodfellow, 2016*). *Chollet & Chollet (2021)* suggest an approach involving the deliberate construction of a complex network, followed by applying regularization techniques to counter overfitting. However, due to the substantial complexity of datasets and models, an alternative strategy of concurrently training multiple models with varied capacities and regularization methods is employed in this study. Weight decay, a regularization technique imposing a penalty function on complexity, reduces the network's tendency to overfit (*Houssein et al., 2024*; *Loshchilov & Hutter, 2016*). $L^2$ weight decay, a common form, adjusts the relative contribution of the norm penalty function through a weight parameter. Finding an optimal alpha value (determining the degree of regularization) necessitates experimentation tailored to the specific situation (*Sterkenburg, 2025*; *Hastie, Tibshirani & Friedman, 2009*). Initial attempts to implement $L^2$ weight decay into the network using specific values encountered technical challenges, causing conflicts within TensorFlow and subsequent crashes (*Jang, 2025*; *Pang, Nijkamp & Wu, 2020*; *Hekler et al., 2019*; *Hastie, Tibshirani & Friedman, 2009*). As a result, prioritizing other aspects of the study overcame these obstacles.

## Metrics

When assessing the performance of a model in a classification task, a frequently employed and straightforward metric is prediction accuracy (*Mohammed & Meira, 2020*; *Davis & Goadrich, 2006*). This metric, as defined by *Mohammed & Meira (2020)*, serves as a common evaluation criterion:

$$Accuracy = \frac{1}{n}\sum_{i=1}^{n} I(y_i = \widehat{y}_i). \tag{1}$$

In a scenario with '$n$' observations, accuracy measures how well a model's estimates match the actual answers. Yet, accuracy can fall short for imbalanced datasets, where it may inflate due to a majority class bias, rendering them less useful (*Liaw et al., 2025*; *He & Garcia, 2009*). To counter this, the receiver operating characteristic curve (ROC)/area under curve (AUC) analysis proves valuable, unaffected by such imbalances (*Ozel et al., 2025*; *Ha, Liu & Liu, 2020*; *Esteva et al., 2019*; *Fawcett, 2006*). Additionally, the connection between CNN outputs and ROC thresholds has made ROC analysis prevalent in evaluating skin lesion CNNs. ROC analysis hinges on the true positive rate (TPR) and false positive rate (FPR), explained through confusion matrices (*Gouda et al., 2022*; *Yap, Yolland & Tschandl, 2018*). In a binary classification scenario with '$n$' observations, '$y_i$' represents true

labels, and '$\widehat{y_i}$' signifies estimated labels for each observation indexed from 1 to '$n$', distinguishing positive ('$c_1$') and negative ('$c_2$') classes (*Hernández-Pérez et al., 2024*; *Hekler et al., 2019*; *Kassem et al., 2021*).

Hence, in line with commonly used metrics in skin lesion classification research, as demonstrated in *Houssein et al. (2024)*, *Ali et al. (2022)*, *Yap, Yolland & Tschandl (2018)*, *Esteva et al. (2019)*, we employ AUC-ROC, TPR, and FPR for performance evaluation.

## TRAINING A CNN

This section outlines the prevalent strategy of employing a pre-trained network derived from the ImageNet challenge as the foundation for training a CNN aimed at skin lesion classification (*Reddy et al., 2023*; *Ha, Liu & Liu, 2020*). This method finds frequent application in research articles (refer to Table 1) and has been proven instrumental in securing victory in a significant Kaggle competition (*Banachewicz & Massaron, 2022*). It is endorsed in dedicated deep-learning literature (*Hernández-Pérez et al., 2024*; *Ali et al., 2022*). Subsequent sections delineate endeavors in constructing and training CNN.

### Preliminary training approach

The primary objective was to train a CNN using the EfficientNetB3 architecture (*Alhichri et al., 2021*), with a Flattening layer followed by three Dense layers. The goal was to explore a large hyperparameter space that extended beyond the scope outlined in "Empirical study". A full search, excluding learning rate tuning, would require training 1,024 models, which was impractical. To address this, the Keras tuner package Hyperband (*Li et al., 2018*) was used, applying a multi-armed bandit strategy to systematically evaluate models within the hyperparameter space (*Ali et al., 2022*). Hyperband's heuristic approach creates a subset of models, trains them briefly for a few epochs, saves the models, and discards inferior performers (*Alhichri et al., 2021*; *Hekler et al., 2019*). The top-performing models are iteratively refined through further training, allowing resources to focus on the most promising candidates (*Chen et al., 2024*; *Li et al., 2018*). A preliminary test with Hyperband used 320 training images (less than 1% of the total 50,000) and 160 validation images (192 × 158 pixels). Training for 15 epochs took approximately 7 h, during which 90 models were evaluated, identifying the best-performing one. This model's base was unfrozen and fine-tuned with 10 additional epochs. As shown in Table 5, the final model had more parameters in the Dense layers, driven by the Flattening layer's large output size.

However, four issues emerged during the evaluation:

1) **Image dimension error:** Images were incorrectly encoded as (192,158) instead of (128,192), affecting performance (*Goodfellow, 2016*).

2) **Runtime limitations:** Testing on a cloud setup led to suboptimal node utilization, with training marked by performance spikes and inefficiencies. Enlarging the images to (512,768) increased the epoch runtime by 12×, rendering a full-scale test infeasible (*Chollet & Chollet, 2021*).

**Table 5 Model summary of the best model.**

| Layer | Output shape | Number of parameters |
|---|---|---|
| EfficientNetB3 (functional) | (None, 6, 5, 1,536) | 10,783,535 |
| Flatten | (None, 46,080) | 0 |
| Dropout | (None, 46,080) | 0 |
| Dense1 | (None, 256) | 11,796,736 |
| BatchNorm | (None, 256) | 1,024 |
| Dense2 (Dense) | (None, 160) | 41,120 |
| Dense (Dense) | (None, 8) | 1,288 |

3) **Dataset inconsistency:** Initially, only the HAM10000 subset of the 2019 dataset was downloaded, encoding eight classes instead of nine. The complete dataset was later obtained and re-encoded to meet the required specifications (*Tschandl et al., 2019*).

4) **Prediction imbalance:** Despite achieving low loss values, the model's predictions overestimated prevalent classes while assigning low probabilities to less common ones, limiting its generalization (*Bria, Marrocco & Tortorella, 2020*).

## Optimizing the training strategy

The initial strategy faced obstacles that required significant time and effort to resolve. An analysis of the winning solution from the 2019 and 2020 Kaggle competition (*GitHub, 2024a, 2024b; Lin et al., 2017*) revealed available code on GitHub (*GitHub, 2024a, 2024b*), but the training process demanded extensive computational resources, exceeding our infrastructure. Additional exploration uncovered contributions from a Kaggle grandmaster, who utilized TPUs—resources not available to us (*Banachewicz & Massaron, 2022*). Another submission by *Jang (2025)* provided valuable insights but also relied on TPUs, requiring image rescaling to a 1:1 ratio. We evaluated this model after resizing images to (256,256), achieving an AUC score of 0.773. Through iterative reviews and small-scale testing, we identified promising adjustments that improved training by focusing on image features rather than class distributions. Key observations included:

- Implementation of GlobalAveragePooling instead of Flattening layers between the base and head, substantially enhancing performance while reducing the parameters in the initial dense layer (*Ali et al., 2022; Minderer et al., 2022*).

- Experimentation with binary-labeled training sets, inspired by Tensorflow (*Jang, 2025; Pang, Nijkamp & Wu, 2020; Dillon et al., 2017*), demonstrated quick and satisfactory results despite potential information loss compared to multilabel training (*Houssein et al., 2024; Bria, Marrocco & Tortorella, 2020*).

- Adoption of a learning rate schedule aligned with *Chollet & Chollet*'s *(2021)* recommendation, diverging from the reviewed models' approaches, showcased substantial performance enhancements (*Ha, Liu & Liu, 2020; Loshchilov & Hutter, 2016*).

- Construction of models with shallow heads akin to Tensorflow (*Jang, 2025*; *Pang, Nijkamp & Wu, 2020*; *Dillon et al., 2017*) designs resulted in significantly fewer parameters than conventionally proposed by *Chollet & Chollet (2021)*.
- Integration of class weights into the optimizer, following the Tensorflow approach (*Pang, Nijkamp & Wu, 2020*; *Dillon et al., 2017*), counteracted skewed class distributions, notably improving model predictions (*Lin et al., 2017*).

The collective impact of these methodologies substantially enhanced our models' capability to discern image features rather than relying solely on class distribution information (*Hernández-Pérez et al., 2024*). Consolidating these strategies, a new model is crafted, expected to surpass the evaluation set performance achieved by the Tensorflow model (*Pang, Nijkamp & Wu, 2020*; *Dillon et al., 2017*).

### Final model training and evaluation

The final configuration focused on three key parameters: label encoding (binary/ multiclass), model head capacity (shallow/deep), and dropout layer strength (0/0.2/0.4/ 0.6). Initially, we attempted to use two dropout sizes (0/0.5) and an L2 kernel regularizer in the first Dense layer. However, this setup caused compatibility issues with EfficientNetB3, leading to recurring errors. As L2 regularization was absent in similar models, we shifted focus to exploring more dropout configurations. A total of 16 models were trained with varying parameter combinations.

To balance convergence and training time, we set the maximum epoch limit to 50 with an early stopping callback (patience = 7). The best model, based on validation loss, was retained. Early stopping minimized computational overhead by halting training when no improvement was observed over seven consecutive epochs (*Caruana & Niculescu-Mizil, 2006*).

While EfficientNet models can handle higher resolutions, they increase computational demands without guaranteed performance improvements (*Tan & Le, 2019*). Thus, we downscaled all images to (256,384) for training, validation, testing, and evaluation. Initial trials showed each epoch took just over two hours. Following prior studies (*Ali et al., 2022*; *Shaikh & Khoja, 2013*), we estimated 35–40 h of total training time per model over 15 epochs.

To speed up training, we ran models in parallel, grouped by target label and model capacity. Four separate scripts were developed, each using one of the four dropout configurations. These scripts applied identical preprocessing steps with an ImageDataGenerator to resize and augment images (*Chollet & Chollet, 2021*). Training batches were set to 64 to ensure the representation of smaller categories, while test batches used a size of 1 to maintain image order alignment with predictions. Each script's total execution time ranged between 8 and 12 days.

## BASELINE EXPERIMENTS

This section outlines three key aspects. Initially, the performance of 69 relevant individuals on the evaluation dataset is detailed, and their average performance establishes a human

baseline. Subsequently, the selection process for determining the CNN baseline among 16 trained CNNs is presented. Finally, an evaluation and comparison of this CNN baseline against both human performances and the Tensorflow (*Pang, Nijkamp & Wu, 2020*; *Dillon et al., 2017*) model is provided.

The performance of each individual is quantified as a pair of FPR and TPR values. The FPR is calculated as the ratio of false positives to the total number of actual negatives, while the TPR, also known as Sensitivity or Recall, is the ratio of true positives to the total number of actual positives (*Hanley & McNeil, 1982*). These metrics are essential for evaluating model performance, especially in medical imaging contexts where class imbalances can significantly impact results.

Establishing a human baseline is critical for understanding the performance of automated systems in clinical settings. Previous studies in dermatology have emphasized the variability of human diagnostic capabilities, often indicating that dermatologists can achieve TPRs in the range of 0.70 to 0.85 (*Rawat, Rajendran & Sikarwar, 2025*; *Khan et al., 2024*). The selection process for identifying the CNN baseline among the 16 trained models should involve comparing metrics like AUC-ROC and cross-validation performance to ensure robustness (*Rawat, Rajendran & Sikarwar, 2025*; *Ophir et al., 1991*). Comparing the CNN baseline to human performance allows for assessing the effectiveness of the model in mimicking or surpassing human diagnostic capabilities (*Ali et al., 2022*; *Esteva et al., 2017*).

Figure 1 illustrates the performance of individual humans, accompanied by an average FPR and TPR denoted by a blue dot. This average value does not signify an ensemble value but represents the mean performance level of humans. It serves as a benchmark for human performance for subsequent analysis and comparison. The average FPR stands at 0.196, while the average TPR is 0.765 (*Grzybowski, Jin & Wu, 2024*).

All 16 trained models, as expounded, undergo evaluation using the ROC/AUC metrics stipulated in "Empirical Study" across training, validation, test, and evaluation datasets. The AUC scores for these models are tabulated in Table 6, with the highest scores per category highlighted in boldface. An examination of the table reveals pertinent observations:

1) **Overfitting indicators:** Scores typically peak on the training data, a common occurrence in machine learning owing to model feature acquisition from the training data. Substantial disparities between training and validation/test scores could signal overfitting (*Khan et al., 2025*; *Goodfellow, 2016*). Test scores tend to be marginally lower than validation scores. Two plausible explanations arise. Firstly, dissimilar incidence rates between the training set (derived from the 2019 Kaggle competition) and the test set could lead to biased performance if the model learned this incidence distribution. Similarly, the evaluation method's difference, where the built-in AUC measure for the Keras package assesses AUC scores based on all prediction values rather than solely the "MEL" category, might inflate train and validation scores.

2) **Dataset quality and incidence rates:** Evaluation data scores fall below test scores, potentially due to qualitative differences in datasets. The Kaggle datasets are thoroughly

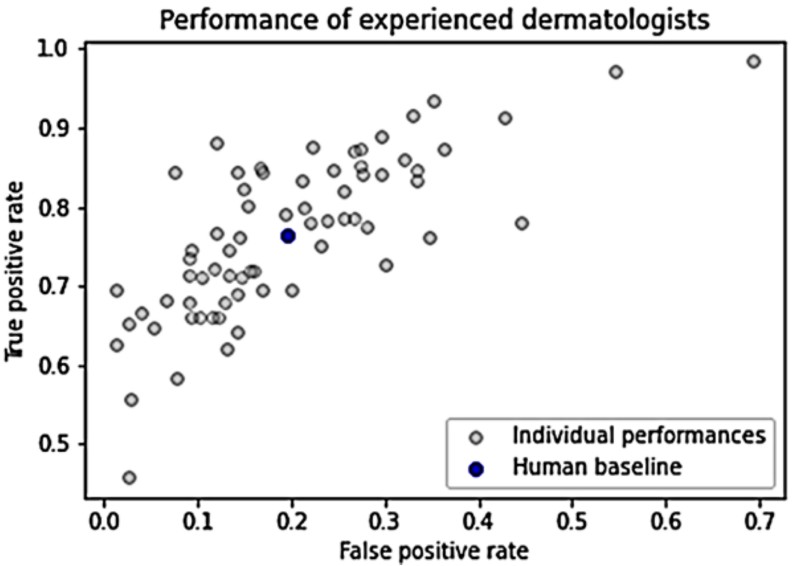

Figure 1  Dermatologists' performance.     

Table 6  Overview of the different models.

| ID | Labels | Depth | Dropout | Train_AUC | Val_AUC | Test_AUC | Eval_AUC |
|---|---|---|---|---|---|---|---|
| 1. | Binary | Deep | 0 | **0.995** | 0.940 | 0.868 | 0.806 |
| 2. | Binary | Deep | 0.2 | 0.99 | 0.948 | 0.895 | 0.766 |
| 3. | Binary | Deep | 0.4 | 0.987 | 0.931 | 0.807 | 0.801 |
| 4. | Binary | Deep | 0.6 | 0.988 | 0.946 | 0.883 | 0.746 |
| 5. | Binary | Shallow | 0 | 0.971 | 0.912 | 0.751 | 0.705 |
| 6. | Binary | Shallow | 0.2 | 0.990 | 0.940 | 0.824 | 0.799 |
| 7. | Binary | Shallow | 0.4 | 0.994 | 0.950 | 0.885 | 0.763 |
| 8. | Binary | Deep | 0.6 | 0.990 | 0.950 | 0.886 | 0.795 |
| 9. | Multiclass | Deep | 0 | 0.990 | 0.940 | 0.887 | 0.809 |
| 10. | Multiclass | Deep | 0.2 | 0.986 | **0.983** | 0.549 | 0.593 |
| 11. | Multiclass | Deep | 0.4 | 0.984 | 0.690 | **0.897** | **0.822** |
| 12. | Multiclass | Deep | 0.6 | 0.981 | 0.981 | 0.885 | 0.762 |
| 13. | Multiclass | Shallow | 0 | 0.981 | 0.927 | 0.766 | 0.762 |
| 14. | Multiclass | Shallow | 0.2 | 0.978 | 0.934 | 0.713 | 0.747 |
| 15. | Multiclass | Shallow | 0.4 | 0.986 | 0.981 | 0.863 | 0.737 |
| 16. | Multiclass | Shallow | 0.6 | 0.983 | 0.980 | 0.886 | 0.737 |

**Note:**
The bold text indicates the highest score per category.

processed, potentially possessing higher quality than the evaluation images (*Banachewicz & Massaron, 2022*; *Khan et al., 2022*; *Esteva et al., 2019*). Additionally, the test dataset might bear a closer resemblance to the training images compared to the evaluation set, causing bias favoring test set performance. Furthermore, differences in incidence rates could contribute to this discrepancy.

3) Model 10 displays anomalous behavior. While training performance is robust, the remaining scores notably plummet. Examination of probability outcomes reveals similarities to the initial model. Visual inspection of AUC progression and loss during training indicates normal initial training, followed by a severe decline in performance—an occasional occurrence possibly stemming from stochastic weight initialization (*Sutskever et al., 2013*). The primary criterion for model selection centers on performance with novel data, specifically test and evaluation set performances. Under these criteria, Model 11 emerges as the superior performer. Notably, it demonstrates optimal performance on novel data while mirroring performance consistency across training and validation sets, indicating an absence of overfitting.

Visual inspection of TensorBoard output underscores a model rapidly learning training data but stabilizing after approximately 17 epochs, exhibiting consistent performance on both the AUC score and loss function between training and validation data. Thus, this model is deemed the baseline CNN model for this study.

On the test set, the baseline CNN attains:

- **AUC score:** AUC score of 0.897, placing it at the $37^{th}$ percentile from the bottom, considerably distant from the winning score of 0.949 in the Kaggle competition field.
- **Evaluation dataset score:** AUC score of 0.822 on the evaluation dataset, surpassing the 0.773 scored by Tensorflow model.

Nonetheless, compared to human baseline performances, the CNN baseline falls short.

- **TPR comparison:** At comparable FPR rates, the average human exhibits TPR = 0.765, whereas the CNN records TPR = 0.694.
- **FPR comparison:** At analogous TPR rates, the average human demonstrates FPR = 0.196, while the CNN displays FPR = 0.320.

A visual comparison of these CNN models and the human average is depicted in Fig. 2. A juxtaposition between the final model and the initial training outcomes demonstrates a substantial enhancement in model performance and behavior. Notably, the final model comprises only 201,188 parameters, significantly fewer than the 11,840,195 parameters in the initial training model, as evident in Table 7. Additionally, Table 8 shows a broader range of prediction values across almost all categories in the final model. This discrepancy signifies the final model's capability to discern image features rather than merely learning the training distribution.

In summary, the baseline CNN converges effectively, outperforming the model (*Pang, Nijkamp & Wu, 2020*) but falling short of the average human performance in the study. The subsequent section will introduce and evaluate hybrid algorithms founded on predictions from both the human and CNN baselines.

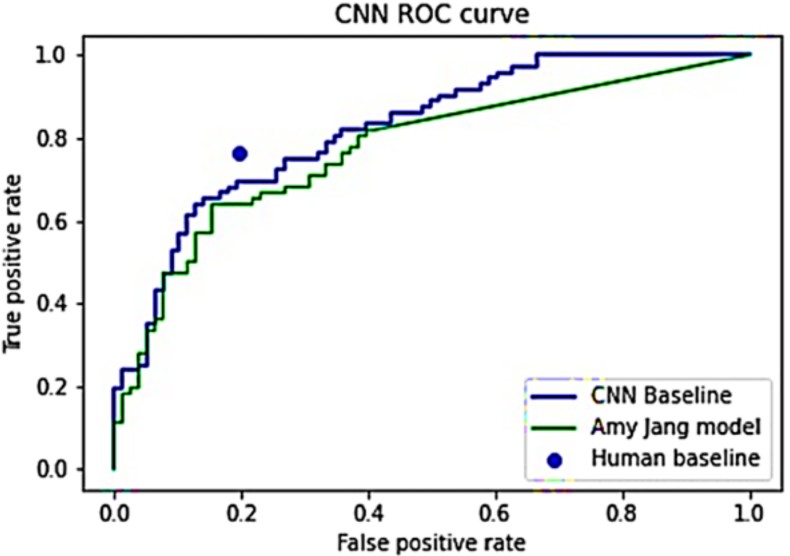

**Figure 2 ROC curves for *Jang*'s *(2025)* model and CNN baseline model.**

**Table 7 Model summary of the top-performing model.**

| Layer | Output shape | Number of parameters |
|---|---|---|
| EfficientNetb3 (Functional) | (None, 1,536) | 10,783,535 |
| dropout (Dropout) | (None, 1,536) | 0 |
| dense 6 (Dense) | (None, 128) | 196,763 |
| dense 7 (Dense) | (None, 128) | 4,128 |
| dense 8 (Dense) | (None, 9) | 297 |

**Table 8 Descriptive statistics of the probability estimates from the top-performing model on the test set.**

| Label | 0 | 1 | 2 | 3 | 4 | 5 | 6 | 7 | 8 |
|---|---|---|---|---|---|---|---|---|---|
| Min | 4.5e−11 | 9.3e−12 | 3.8e−09 | 2.6e−11 | 3.9e−07 | 2.0e−6 | 1.4e−11 | 7.5e−08 | 1.1e−11 |
| Median | 5.5e−06 | 9.3e−12 | 6.1e−05 | 2.3e−06 | 6.6e−04 | 3.9e−3 | 1.8e−06 | 0.99 | 1.8e−06 |
| Max | 0.94 | 0.59 | 0.15 | 0.75 | 0.996 | 1.0 | 0.29 | 1.0 | 0.15 |

# HYBRID ALGORITHMS

## Augmented hybrid approach

In certain instances, image classification poses varying levels of difficulty, presenting disparities between human and machine capacities (*Kaluarachchi, Reis & Nanayakkara, 2021*). The study by *Han et al. (2018)* demonstrated discrepancies in image interpretation, where image challenges for human subjects exhibited notably high performance when processed by algorithms. Similarly, *Marchetti et al. (2020)* revealed enhancements in performance by replacing uncertain human responses with predictions generated by a

CNN. Defining a mutual relationship in the difficulty of image interpretation—where some images present challenges for humans but not for machines, while others are easily interpreted by humans yet pose complexity for computational systems (*Archana & Jeevaraj, 2024*). Moreover, it proposes leveraging CNN prediction values as indicators to delineate images deemed "easy" or "difficult" by the system (*Sun et al., 2023*). If substantiated, this insight could facilitate the construction of a synthesized list comprising both human and computer predictions, derived from the certainties inherent in CNN predictions (*Jackson et al., 2025*).

### Algorithm

In the scenario presented in Eqs. (2)–(7), there exists an array denoted as $A$, comprising prediction values $a_1$ through $a_n$, where '$n$' represents the count of assessed images. The task involves arranging this array in ascending order, leading to the formation of array $B$, consisting of elements $b_1$ through $b_n$. Simultaneously, it is imperative to maintain a clear correspondence or mapping between the index values of the original array $A$ and the resulting array $B$. Introducing a parameter labeled as '$s$', let us define '$i$' and '$j$' as follows:

$$i = \left| \frac{n}{s} \right| \tag{2}$$

$$j = \left| \frac{(s-1)n}{s} \right|. \tag{3}$$

The constraint stipulates that '$s$' must be greater than 2 to uphold the condition where '$i$' is less than '$j$'. Subsequently, the partitioning of '$B$' into three lists can be executed as:

$$B_{lower} = [b_1, b_2, \ldots, b_i] \tag{4}$$

$$B_{inner} = [b_{i+1}, b_{i+2}, \ldots, b_j] \tag{5}$$

$$B_{upper} = [b_{j+1}, b_{j+2}, \ldots, b_n]. \tag{6}$$

The merging of $B_{lower}$ and $B_{upper}$ is designated as $B_{outer}$.

$$B_{outer} = [b_1, b_2, \ldots, b_i, b_{j+1}, b_{j+2}, \ldots, b_n]. \tag{7}$$

The array denotes values extracted from $B$ that demonstrate heightened "certainty" by their proximity to either 0 or 1. Conversely, the $B_{inner}$ signifies values from $B$ characterized by reduced certainty, specifically those closer to 0.5. Upon the establishment of both the internal and external lists, the indices corresponding to the images within one of these lists can be utilized as inputs for Algorithm 1, designated as the "Subset index list". Subsequently, this algorithm substitutes the CNN predictions linked to the Subset index list with human responses randomly chosen for those specific images (lines 4–11), thereby generating a substituted predictions list.

Following this, an analysis employing standard ROC/AUC methodology, as introduced in "Empirical Study", is conducted on this list. Due to the algorithm's random selection of human predictions for each index, stochasticity becomes inherent, resulting in varying

| Algorithm 1 Augmented hybrid approach. |
|---|

**START**

**input:** *CNN predictions, Human predictions, Ground truths, Subset index list, Number of iterations, threshold list*

**output**: *ROC Curve, AUC score*

1 *ROC_Curves = 3DTensor*

2 *AUC_Scores = list*

3 **for** *each iteration* **do**

4     *Create substituted prediction list*:

5     *sub_predictions = list*

6     **for** *each index in CNN prediction* **do**

7         **if** *index in Subset index list* **then**

8             *Pick random human prediction for corresponding image*

9             *sub_predictions[index] ← humanpredictions[index]*

10         **else**

11             *sub_predictions[index] ← CNNpredictions[index]*

12     **for loop end**

13 *Create ROC Curves & AUC Scores*:

14 *ROC, AUC ← ROC/AUCAnalysis(input = (sub_predictions, Groundtruths, thresholdlist)*

15 *UpdateROC_Curves ← ROC (stacking the dataframes on top of each other)*

16 *UpdateAUC_scores ← AUC*

17 **for loop end**

18 *Generate average ROC Curve*:

19 **for** *each cell $c_{ijn}$ in ROC_curve* **do**

20 $\left| c_{ij} \leftarrow \frac{1}{n} \sum_{n=1}^{n} c_{ijn} \right.$

21 **for loop end**

22 *Plot = Lineplot showing Sensitivity and Specificity measures*

23 *Average AUC ← $\frac{1}{n} \sum_{n=1}^{n} AUC\_scores$*

24 *Return Plot, Average AUC*

**END**

outcomes across different trials. Addressing this variability involves executing the setup multiple times, as per the tenet of the strong law of large numbers, which asserts that the anticipated result from an infinite number of trials will converge toward the population parameter. In the algorithm's specified lines (18–23), a sequence of 1,000 trials was executed, collecting ROC curves and AUC scores for each trial. These values were subsequently consolidated using averaging. The method employed for averaging involved

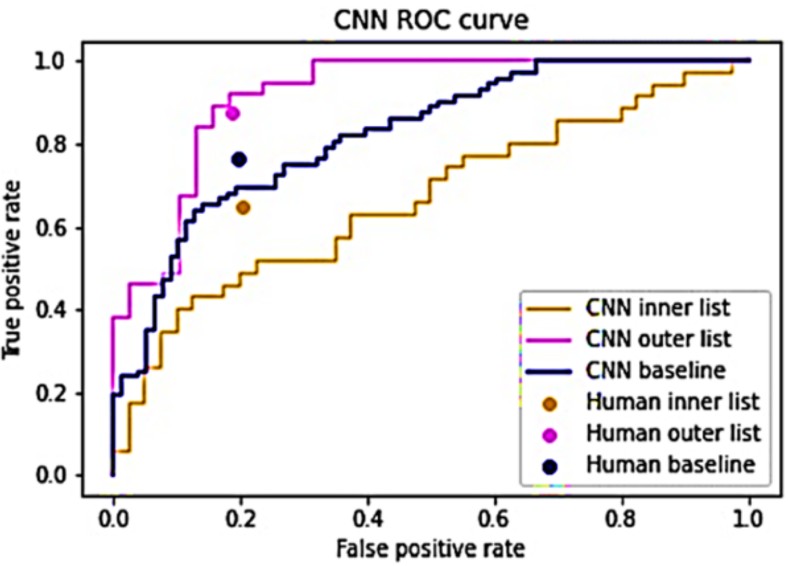

**Figure 3** **Performance of humans and CNN for the inner and outer sub-lists.**

treating individual ROC curves as distinct DataFrames, each composed of (FPR and TPR) pairs. These DataFrames were concatenated vertically, forming a 3D Tensor structure. In this arrangement, each cell could be denoted as $c_{ijn}$, where '$i$' signifies the row index, '$j$' represents the column index, and '$n$' indicates the depth corresponding to the specific iteration number. The subsequent step involved computing the average values across the depth dimension, resulting in the derivation of a final DataFrame.

### *Testing differences between humans and CNNs*

The hybrid algorithm operates under the premise that humans and CNNs exhibit varying proficiency in analyzing distinct images, with their performances showing minimal correlation (*Bozkurt, 2022*). A strong correlation exists between their performances, substituting one entity's prediction with the other should yield negligible or no impact. To validate this presumption, the CNN predictions were segregated into inner and outer lists following Eqs. (5) and (7). A value of $s = 4$ was employed to maintain equal list sizes (*Wang, Wong & Lu, 2020*; *Fawcett, 2006*). Both lists underwent evaluation by both the baseline CNN model and human evaluators. The outcomes, depicted in Fig. 3, highlight differential ease in identifying certain images. The CNN's performance appears least optimal for the inner list, registering an AUC of 0.664, while demonstrating superior performance on the outer list, yielding an AUC of 0.917. Similarly, human evaluators displayed lower performance on the inner list and higher performance on the outer list. Notably, their TPR exhibited variation between the lists, whereas the FPR remained relatively consistent. This substantiates the assertion that certain images pose greater classification challenges while suggesting the feasibility of utilizing CNN predictions to specifically target these challenging images (*Müller et al., 2024*). For the inner list, CNN's performance significantly trails behind human performance. Conversely, on the outer list, CNN's performance marginally surpasses human performance (*Mahmood et al., 2024*).

These findings corroborate the notion that humans and CNNs encounter difficulties in analyzing dissimilar images, thereby supporting the argument against a strong correlation between their performances (*Mahmood et al., 2024*).

### Augmented hybrid results

After demonstrating variations in image difficulty for classification and the lack of a strong correlation between human and CNN performance, the subsequent phase involves ensembling human and CNN responses into a unified list (*Ganaie et al., 2022*; *Wu et al., 2022*). It is postulated that an arrangement where human answers constitute the inner elements and CNN predictions form the outer elements will outperform the baseline performances (*Akram et al., 2025*; *Archana & Jeevaraj, 2024*). To evaluate this hypothesis, Algorithm 1 is executed twice: once with the inner list as the "Subset index list" and once with the outer list in the same role. Both instances of the algorithm are run 1,000 times, with an expectation of convergence towards the anticipated ROC curve and AUC values. Figure 4 presents the outcomes derived from generating and scrutinizing the inner and outer substitution lists.

The outer substitution list broadly mirrors the CNN baseline model, deviating slightly at the extremities, showcasing an AUC score of 0.798—marginally lower than the baseline CNN score of 0.822 (*Archana & Jeevaraj, 2024*). In contrast, the inner substitution line exhibits a distinct pattern: below the CNN baseline within FPR [0, 0.17], surpassing the baseline within FPR [0.17, 0.4], and descending below the baseline within FPR [0.4, 1]. Its AUC score of 0.772 falls slightly beneath the CNN baseline. Nevertheless, the inner substitution line boasts a higher TPR (0.782) compared to the human baseline at similar FPR levels, while also demonstrating a lower FPR (0.182) compared to the human baseline at similar TPR values (*Shahid et al., 2025*; *Caruana & Niculescu-Mizil, 2006*).

## STUDY LIMITATIONS

There are certain limitations of this study. Addressing these limitations could strengthen the robustness and applicability of the study's findings in real-world clinical practice.

### Ethical considerations

The ethical considerations around having machine-driven decisions in life-critical medical situations using the pure augmented hybrid algorithm are not fully explored. Granting final decision authority to humans is mentioned as a solution but not fleshed out. This study is the first step toward a hybrid algorithm. Nonetheless, this could be addressed by granting a human final decision-making authority which is aligned with the global concept of human-in-the-loop (*Sasseville et al., 2025*; *Schuitmaker et al., 2025*; *Siddique et al., 2024*; *van den Berg, 2024*; *Topol, 2019*).

### Limited dataset size and generalizability

The study's findings may not be directly applicable to all dermatological settings due to the specific dataset used, which consists of 150 images focused on particular characteristics of skin lesions (*Reddy et al., 2023*; *Shahid et al., 2025*; *Tschandl et al., 2019*). The small size of the evaluation dataset and the potential selection bias limits the generalizability of the

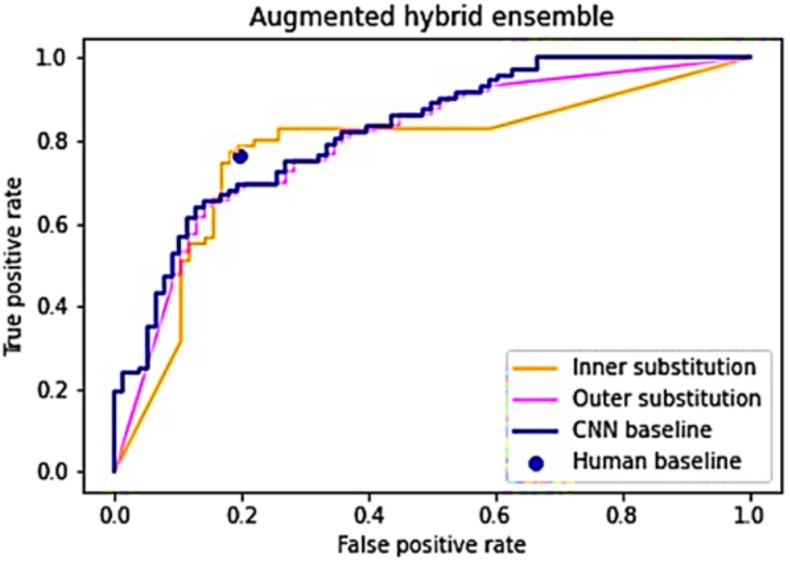

**Figure 4** An average of 1,000 simulations of the inner and outer hybrid lists.

results (*Sabazade et al., 2025*; *Esteva et al., 2017*). The study does not explicitly address whether the dataset fully represents real-world clinical scenarios (*Liu et al., 2024*; *Bejnordi et al., 2017*). Different populations, imaging techniques, or lesion types may yield different results (*Grzybowski, Jin & Wu, 2024*; *Winkler et al., 2019*; *Bejnordi et al., 2017*). A larger, more diverse dataset in future studies would help address these concerns and strengthen the conclusions (*Yan et al., 2025*; *Chen et al., 2024*; *Tschandl et al., 2020*).

## Model complexity and interpretability

While the CNN models used, particularly EfficientNetB3, demonstrate strong performance, they inherently function as "black boxes," limiting interpretability (*Räz, 2024*). This lack of transparency is a critical concern in medical settings, where clinicians require not only accurate predictions but also insights into the reasoning behind them to build trust and validate results. Without interpretability, it becomes difficult to detect biases, troubleshoot errors, or confidently apply the model's predictions in high-stakes scenarios. Incorporating techniques such as saliency maps, attention mechanisms, or SHAP values could enhance transparency by identifying which features or regions of an image influence the model's output (*Cohen-Inger et al., 2025*; *Deng et al., 2024*; *Soomro, Niaz & Choi, 2024*). Hybrid approaches, combining interpretable rule-based models with deep learning, may also strike a balance between performance and explainability (*Khalil et al., 2023*; *Caruana & Niculescu-Mizil, 2006*). While this study focuses primarily on performance, interpretability remains a crucial area for future research to ensure reliable clinical integration (*Coots et al., 2025*; *Baumann et al., 2024*). Addressing the trade-off between model complexity and interpretability will be key to gaining practitioner trust and achieving better patient outcomes (*Guyton, Pak & Rovira, 2025*; *Bria, Marrocco & Tortorella, 2020*).

### Inclusion of metadata

The decision to exclude metadata from the model may overlook potentially valuable information that could improve diagnostic accuracy (*Khan et al., 2025*; *Duan et al., 2024*; *Wang, Wong & Lu, 2020*). Incorporating metadata, such as patient age and lesion location, might enhance the model's performance (*Guermazi et al., 2024*; *Kania, Montecinos & Goldberg, 2024*; *Esteva et al., 2019*).

### Resource intensive training

Training deep learning models, especially on large datasets with complex architectures, requires significant computational resources and time (*Rahman et al., 2021*; *Bria, Marrocco & Tortorella, 2020*; *Litjens et al., 2017*). This limitation could hinder the broader adoption and replication of the study's findings, particularly in resource-constrained settings (*Zhang et al., 2024a*, *2024b*; *Topol, 2019*).

### Dependency on histopathological examination

The ground truth labels for the dataset were established *via* histopathological examination, which itself has limitations (*McCaffrey et al., 2024*; *Göndöcs & Dörfler, 2024*), including sampling bias (*Wang et al., 2024*; *Webb et al., 2024*) and interobserver variability (*Pinello et al., 2025*; *Shinde et al., 2025*). Reliance solely on histopathology may introduce errors in the dataset labels (*Zhang et al., 2024b*).

### Human baseline variability

The human baseline performance was established using a subset of participants experienced in dermoscopy. However, individual variability in human performance (*Naseri & Safaei, 2025*; *Stevens et al., 2025*; *Naeem et al., 2024*), even among experienced dermatologists (*Gupta et al., 2025*; *Rubegni et al., 2024*), could introduce uncertainty (*Sanz-Motilva et al., 2024*) in comparison with the CNN models (*Miller et al., 2024*; *Ali et al., 2023*).

### Dataset imbalance

The dataset used for training and evaluation may suffer from class imbalance issues (*Liu et al., 2020*; *He & Garcia, 2009*), particularly with the low incidence rate of malignant cases (*Gurcan & Soylu, 2024*). This imbalance could affect the model's performance and generalizability (*Fang et al., 2025*).

### Exploring model enhancements for improved performance

This study demonstrates how a hybrid approach—combining CNN predictions with human expertise—outperforms individual baselines (*Nugroho, Ardiyanto & Nugroho, 2023*), including both standalone CNNs (*Liu et al., 2025*) and human assessments (*Selvaraj et al., 2024*; *Esteva et al., 2017*). However, the AUC scores achieved are lower than those reported in recent studies, such as *Houssein et al. (2024)* and *Nugroho, Ardiyanto & Nugroho (2023)*, which utilize more advanced CNN architectures and training techniques.

Future work will focus on testing our hybrid approach with these newer methods and exploring ways to adapt or incorporate them into our framework to achieve further

performance gains. Benchmarking against these recent advancements will help ensure our approach remains competitive and aligned with the latest developments in skin lesion classification.

## CONCLUSION

This research introduces a novel augmented hybrid approach that combines the strengths of CNNs with selective human intervention, aimed at enhancing skin lesion classification accuracy. By leveraging the EfficientNetB3 backbone, known for its balance between performance and efficiency, this study advances the field of medical image analysis with a focus on practicality and scalability (*Esteva et al., 2019*). The hybrid algorithm prioritizes high-confidence CNN predictions while delegating uncertain cases to medical experts, thereby optimizing diagnostic outcomes with minimal human resource expenditure.

Our comprehensive evaluation of the ISIC-2019 and ISIC-2020 datasets compared against 69 trained medical professionals demonstrates the promise of this approach (*ISIC, 2024*). The baseline CNN model achieved a competitive AUC score of 0.822, performing close to human experts. However, the hybrid model improved upon these results, achieving a TPR of 0.782 and reducing the FPR to 0.182, showcasing the effectiveness of combining human and machine intelligence. These findings underscore the practical potential of integrating CNNs into clinical workflows while ensuring that human expertise remains central to decision-making (*Rawat, Rajendran & Sikarwar, 2025*; *Gholizadeh, Rokni & Babaei, 2024*).

While the hybrid approach offers improved diagnostic accuracy and resource efficiency, challenges persist. Issues such as dataset imbalance, model interpretability, and computational resource demands highlight the need for further research to refine and generalize the methodology (*Strika et al., 2025*; *Char, Shah & Magnus, 2018*). The exclusion of metadata, though intentional in this study, also points to opportunities for future work that may enhance diagnostic performance by incorporating contextual clinical information (*Hermosilla et al., 2024*; *Jones et al., 2022*). Moreover, ethical considerations surrounding human-in-the-loop frameworks require careful attention to ensure that the technology serves as a support system, not a replacement, for clinical judgment (*Lee et al., 2025*).

This research contributes to the growing body of literature on AI-assisted diagnostics by demonstrating the potential of hybrid intelligence models to bridge the gap between human expertise and algorithmic efficiency. The results indicate that well-structured collaboration between CNNs and medical professionals can mitigate the limitations of both systems. Moving forward, this hybrid framework offers a scalable, pragmatic solution for clinical settings, fostering more reliable and accurate skin lesion diagnosis while efficiently managing healthcare resources.

### Funding
The authors received no funding for this work.

## Competing Interests

Deep Himmatbhai Ajabani is employed by Source InfoTech Inc. and Karar Ali is employed by VentureDive Pvt. Limited.

## Author Contributions

- Deep Ajabani conceived and designed the experiments, performed the experiments, analyzed the data, performed the computation work, prepared figures and/or tables, authored or reviewed drafts of the article, and approved the final draft.
- Zaffar Ahmed Shaikh conceived and designed the experiments, performed the experiments, analyzed the data, performed the computation work, prepared figures and/or tables, authored or reviewed drafts of the article, and approved the final draft.
- Amr Yousef conceived and designed the experiments, performed the experiments, analyzed the data, performed the computation work, prepared figures and/or tables, authored or reviewed drafts of the article, and approved the final draft.
- Karar Ali performed the experiments, analyzed the data, performed the computation work, prepared figures and/or tables, authored or reviewed drafts of the article, and approved the final draft.
- Marwan A. Albahar analyzed the data, performed the computation work, prepared figures and/or tables, authored or reviewed drafts of the article, and approved the final draft.

## Data Availability

The ISIC-2019 and ISIC-2020 datasets are available at: https://www.isic-archive.com/.

The winning solution (*i.e.*, code and algorithm) of the 2019 SIIM-ISIC melanoma classification challenge (*Ha, Liu & Liu, 2020*) is available at GitHub (*GitHub, 2024a*) at https://github.com/haqishen/SIIM-ISIC-Melanoma-Classification-1st-Place-Solution.

The original code/algorithm of the 2020 SIIM-ISIC melanoma classification challenge winner is available on GitHub: https://github.com/ISIC-Research/ADAE.

The BCN20000 dataset of the 19,424 images of skin lesions captured from 2010 to 2016 of the Three-Point Checklist of Dermoscopy (*Combalia et al., 2019*) is available at arXiv: https://doi.org/10.48550/arXiv.1908.02288.

The training data for the ISIC-2019 dataset is available at Kaggle: https://www.kaggle.com/datasets/andrewmvd/isic-2019.

The original melanoma skin cancer dataset of 10,000 images is available at Kaggle: https://www.kaggle.com/datasets/hasnainjaved/melanoma-skin-cancer-dataset-of-10000-images.

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
