# Peer review of "Enhancing skin lesion classification: a CNN approach with human baseline comparison"

_PeerJ Computer Science, doi:10.7717/peerj-cs.2795_

## Round 0.1 · original submission · Major Revisions

Dear authors,

Thank you for submitting your article. Reviewers have now commented on your article and suggest major revisions. We do encourage you to address the concerns and criticisms of the reviewers and resubmit your article once you have updated it accordingly.

When submitting the revised version of your article, it will be better to address the following:

1. Section 3 consists of only one too long paragraph that is hard to read. This paragraph should be divided into two or more.
2. References should be written according to PeerJ referencing style.
3. Figures should be polished.
4. Equations should be used with equation number. Explanation of the equations should be checked. Definitions and boundaries of all variables should be provided. Necessary references should also be given.
5. Reviewer 1 and Reviewer 3 have asked you to provide specific references. You are welcome to add them if you think they are useful and relevant. However, you are under no obligation to include them, and if you do not, it will not affect my decision.

Reviewer 1 ·

Basic reporting

This study presents interesting work on combining CNN and human expertise for skin lesion classification, but there are several areas where the methodology, analysis, and presentation could be strengthened to increase the impact and reliability of the findings.

Experimental design

Major:

1. Fundamentally, this study is short in document length and has few references. Readers do not read papers simply to see algorithm and result metrics. The study should review as many state-of-the-art skin diagnosis CNN models as possible, and express that it has taken a 'step forward' better than all of these based on specific insights. For example, this study lacks techniques proposed in "Ant Colony and Whale Optimization Algorithms Aided by Neural Networks for Optimum Skin Lesion Diagnosis: A Thorough Review"; "SNC_Net: Skin Cancer Detection by Integrating Handcrafted and Deep Learning-Based Features Using Dermoscopy Images"; "Enhancing Skin Lesion Detection: A Multistage Multiclass Convolutional Neural Network-Based Framework"; "Deep Learning-based Classification of Abrasion and Ischemic Diabetic Foot Sore Using Camera-Captured Images".

I have listed only four papers from 2023-2024 that I am familiar with, as my knowledge of skin lesion CNN papers is not exhaustive, but excellent recent CNN papers in 2024 generally compare up to 60-70 papers briefly. Please describe the content in detail and compare all 2023-2024 studies, and prove that your model is SOTA.

Validity of the findings

Minor:

I would like to suggest that the authors address these minor limitations in the article, either by discussing them in the limitations section or, where feasible, by making the appropriate revisions:

1. The study uses a relatively small dataset (150 images) for the final evaluation, which may limit the generalizability of the results. A larger and more diverse dataset would strengthen the conclusions.
The authors do not provide details on how they ensured the evaluation dataset is representative of real-world clinical scenarios. There may be potential selection bias.
The process of selecting the 69 experienced participants for human performance comparison is not thoroughly explained. More details on participant selection criteria and demographics would be helpful.

2. While the authors experiment with different CNN architectures and hyperparameters, they do not provide a clear justification for why EfficientNetB3 was ultimately chosen over other options. A more systematic comparison of different architectures would strengthen the study.
The authors mention using early stopping during training, but do not provide details on other regularization techniques used to prevent overfitting.

3. The study primarily relies on AUC as the performance metric. Including additional metrics like sensitivity, specificity, and F1 score would provide a more comprehensive evaluation.
There is limited discussion on the statistical significance of the performance differences between the CNN model and human experts. Statistical tests would add rigor to the comparisons.

4. While the hybrid approach shows promise, the authors do not thoroughly explore different ways of combining human and CNN predictions. Alternative fusion strategies could potentially yield better results.
The impact of varying the parameter 's' in the hybrid algorithm is not extensively analyzed. A sensitivity analysis of this parameter would be valuable.

5. The authors acknowledge some limitations, but do not discuss potential biases in the dataset or how these might affect the model's real-world performance.
There is limited discussion on the computational resources required for training and deploying the model, which is important for practical implementation.

Additional comments

I acknowledge that the authors have already done commendable work on this study. Thank you for your valuable contributions to our field of research. Since I want to this manuscript to be one of the best PeerJ paper, I look forward to reviewing the revised manuscript.

·

Basic reporting

Accept the paper with Minor Modifications.


The abstract and the paper describe an implemented hybrid algorithm without explicitly detailing its specific combinations. Additionally, keywords requiring modification should focus on essential elements such as algorithm names or key parameters discussed in the paper, rather than broader terms like society or international collaboration.

In the introduction, it's appropriate to outline the research gaps, providing reviewers with a clear idea. In the related works or survey field, incorporating more survey papers can enhance understanding and highlight the relevance of the gaps identified.

Experimental design

Accept the paper with Minor Modifications.

It is advisable to include more mathematical formulas, leveraging the promising results. The paper includes tables and graphs, which complement its findings. Organizing the content with a structured architecture enhances reviewers' comprehension.

Validity of the findings

Accept the paper with Minor Modifications.


The findings are clearly presented, and efforts have been made to refine them for better readability and visual appeal.

Additional comments

Accept the paper with Minor Modifications.

Reviewer 3 ·

Basic reporting

Language: The language used in the article is clear and professional, but some sections could be more concise or clarified to avoid redundancy. For example, some parts of the methodology repeat the same concepts with slight variations, which could be streamlined.

Background: The introduction and background provide sufficient context. However, the article could benefit from further elaboration on the novel contributions of this study, particularly in how the hybrid approach outperforms existing methods.

Structure: The structure generally adheres to journal standards, but some sections could be condensed or split to improve focus. For instance, the "Related Works" section could be more concise, concentrating on studies that are directly relevant and innovative.

Experimental design

Method Details: While the methods are described in detail, some sections could be expanded, such as the rationale behind selecting specific parameters for the CNN model and a more detailed explanation of the data augmentation process. This is crucial for enabling accurate replication.

Data Preprocessing Evaluation: The discussion on data preprocessing is adequate, but it could be enhanced by explaining how this preprocessing influenced the final results and whether additional considerations are needed for different datasets.

Validity of the findings

Significance Discussion: The article could better articulate how the findings contribute to advancements in AI-based dermatology. For instance, it could further explain the real-world application potential and how it addresses limitations of previous research.

Conclusion and Implications: While the conclusions are well stated, they could be strengthened by discussing unanswered questions or limitations that might require further research. Additionally, a deeper discussion on limitations such as the challenges in model interpretability should be included.

Additional comments

General comments: As an idea, combining CNN performance and related physician kinaras is novel. However, the performance does not show impressive results. Some of my suggestion points:

(1) The AUC of your approach is very far below other published researchers such as:

Houssein EH, Abdelkareem DA, Hu G, Hameed MA, Ibrahim IA, Younan M. An effective multiclass skin cancer classification approach based on deep convolutional neural network. Cluster Comput 2024;1. doi:10.1007/s10586-024-04540-1.

Nugroho ES, Ardiyanto I, Nugroho HA. Boosting the performance of pretrained CNN architecture on dermoscopic pigmented skin lesion classification. Ski Res Technol 2023;29(11):1-10. doi:10.1111/srt.13505.

You should cite these two studies as updated results in this research area.

(2) I don't think you have a strong foundation in your choice of CNN backbone. Indeed, in the competition EfficientNetB3 performed well, but did you get the detailed configuration they used such as the hyper-parameters? The difference in the configuration of a model will produce very different results. You should do a comprehensive model selection as was done in the study I mentioned earlier.

(3) I suggest that to improve the performance of the model, you should solve the problem of imbalanced datasets used for model training.

(4) It is recommended that a visual depiction in the form of at least a block diagram be presented in the manuscript to make it easier for readers to understand your proposed model approach.

---

## Round 0.2 · Minor Revisions

Dear Authors,

We are awaiting the submission of the revised manuscript, with the requisite minor edits applied, in accordance with the feedback provided by Reviewer 2.

Best wishes,

Reviewer 1 ·

Basic reporting

All comments have been thoroughly addressed.

Experimental design

All comments have been thoroughly addressed.

Validity of the findings

All comments have been thoroughly addressed.

Additional comments

I extend my gratitude to both the authors and editors for taking my opinions into consideration during the review of this manuscript.

·

Basic reporting

Revised manuscript is better than the previous one. Literature concentration is more but introduction of the contribution also similar like literature. Subjective of the manuscript is clear, it will be more effective if the writing with less usage of every paragraph with citation.

Experimental design

methods and algorithm are clear but the diagrammatic of the approach is good to attention in the research (which is missing)

Validity of the findings

results of the proposed approaches are validated but with less amount of data size.

Additional comments

Still the readability of the paper is not up to the mark, each and every section the literature review or citations are added for every statement.

Reviewer 3 ·

Basic reporting

I appreciate the effort the authors put into addressing the comments from the first round of review. The revisions, particularly in the Introduction and Related Works sections, provide much better clarity and context about the study's contributions. The addition of relevant references and the improved organization of the related works make this section more focused and informative.

The language throughout the manuscript is professional, clear, and no longer redundant as previously noted. Overall, the revisions have met my concerns in this area, and the manuscript now aligns well with journal standards.

Experimental design

The authors have clarified the methodology, particularly regarding the CNN experiments, which are now presented more systematically in Subsections 5.1, 5.2, and 5.3. Their explanation of the grid search and hyperparameter tuning process adds transparency to their approach.

While there are no additional ablation studies on data preprocessing, the justification provided and the references added in Subsection 3.1 are sufficient. I also appreciate the discussion about how preprocessing methods could be explored in future research. With these updates, my concerns about the experimental design have been addressed.

Validity of the findings

The authors have significantly improved the Introduction and Conclusion sections, clearly highlighting the contributions and real-world significance of this study. They have effectively explained the potential applications of their hybrid approach, including how it reduces diagnostic errors and costs.

The addition of Section 8: Study Limitations and Section 8.3: Model Complexity and Interpretability demonstrates a thoughtful reflection on the challenges of their proposed method. These revisions provide a balanced discussion of the study’s findings and future directions, which adequately address my earlier concerns.

Additional comments

I appreciate the authors for incorporating the references I suggested and for revising the manuscript in line with the feedback provided in the first round. The changes have made the study clearer, more focused, and better aligned with recent advancements in the field.

Overall, I am satisfied with the revisions, and the manuscript has addressed all my concerns. I recommend this paper for consideration for publication.

---

## Round 0.3 · accepted · Accept

Dear Authors,

Thank you for addressing the reviewers' comments. Your manuscript now seems sufficiently improved and ready for publication.

Best wishes,

·

Basic reporting

Updated version of the paper presentation is good. updated as per my review remarks given last time

Experimental design

Addressed the technical part of the approach already.

Validity of the findings

no comment'

Additional comments

overall, the updated version much better than previous one the presentation is good now.